# The dynamics of functional brain network segregation in feedback-driven learning
Xiaoyu Wang [1]✉, Katharina Zwosta[1], Julius Hennig [2], Ilka Böhm[2], Stefan Ehrlich[2,3], Uta Wolfensteller[1] & Hannes Ruge [1]

Prior evidence suggests that increasingly efficient task performance in human learning is associated with large scale brain network dynamics. However, the specific nature of this general relationship has remained unclear. Here, we characterize performance improvement during feedback-driven stimulus-response (S-R) learning by learning rate as well as S-R habit strength and test whether and how these two behavioral measures are associated with a functional brain state transition from a more integrated to a more segregated brain state across learning. Capitalizing on two separate fMRI studies using similar but not identical experimental designs, we demonstrate for both studies that a higher learning rate is associated with a more rapid brain network segregation. By contrast, S-R habit strength is not reliably related to changes in brain network segregation. Overall, our current study results highlight the utility of dynamic functional brain state analysis. From a broader perspective taking into account previous study results, our findings align with a framework that conceptualizes brain network segregation as a general feature of processing efficiency not only in feedback-driven learning as in the present study but also in other types of learning and in other task domains.

When humans are learning novel behaviors, they transition from a highly controlled processing mode during initial rule acquisition towards a more automatic processing mode associated with increasingly fluent behavior[1]. In human trial-and-error learning, the initial acquisition phase is dominated by complex strategic rule extraction processes typically reflected by a marked increase in response accuracy followed by a phase of rule consolidation at near-ceiling accuracy levels and decreasing response times[2–4]. These behavioral learning-related changes have been shown to be associated with widespread changes in functional brain organization[2,3,5–7]. In particular, analyses of functional brain network dynamics have revealed that the learning-related transition from controlled to automatic processing modes seems to be associated with increasingly segregated brain networks and a close relationship between network segregation and learning rate, both on a slow timescale[8] and on a fast timescale[7].

An influential theoretical framework has conceptualized the transition from more controlled to more automatic processing modes as a transition from goal-directed to habitual behavior. From this perspective, human behavior is assumed to be initially goal-directed, involving the anticipation of future outcomes, and an association is built between the response and the outcome (R-O) in the respective stimulus context (S). With increasing

practice, however, behavior is assumed to become less and less governed by outcome anticipation and instead becomes more and more driven by stimulus-response (S-R) associations.[9–12]

To better understand how the transition from goal-directed behavior into habitual behavior might be related to inter-individual differences in functional brain organization, our group has previously developed a new experimental paradigm[13] (Fig. 1). Specifically, subjects learned novel stimulus-response (S-R) associations by trial-and-error and practiced these for an extended period of time. This was followed by a test phase during which the monetary outcomes associated with the acquired habitual responses could no longer be obtained. Habit strength was quantified by how strongly the acquired S-R associations interfered with competing goal-directed actions (goal-habit competition test). More recently, we have demonstrated that functional connectivity changes involving the sensorimotor and the cingulo-opercular network (CON) contributed most prominently to habit strength prediction[14]. However, it still remains unclear how quickly goal-directed behavior transitions into habitual behavior for different individuals and, most importantly, how the transition speed is related to large-scale functional brain network dynamics.

[1]Faculty of Psychology, Technische Universität Dresden, Dresden, Germany. [2]Translational Developmental Neuroscience Section, Division of Psychological and Social Medicine and Developmental Neurosciences, Faculty of Medicine, TU Dresden, Dresden, Germany. [3]Eating Disorder Treatment and Research Center, Department of Child and Adolescent Psychiatry, Faculty of Medicine, Technische Universität Dresden, Dresden, Germany. ✉e-mail: xiaoyu.wang3@tu-dresden.de

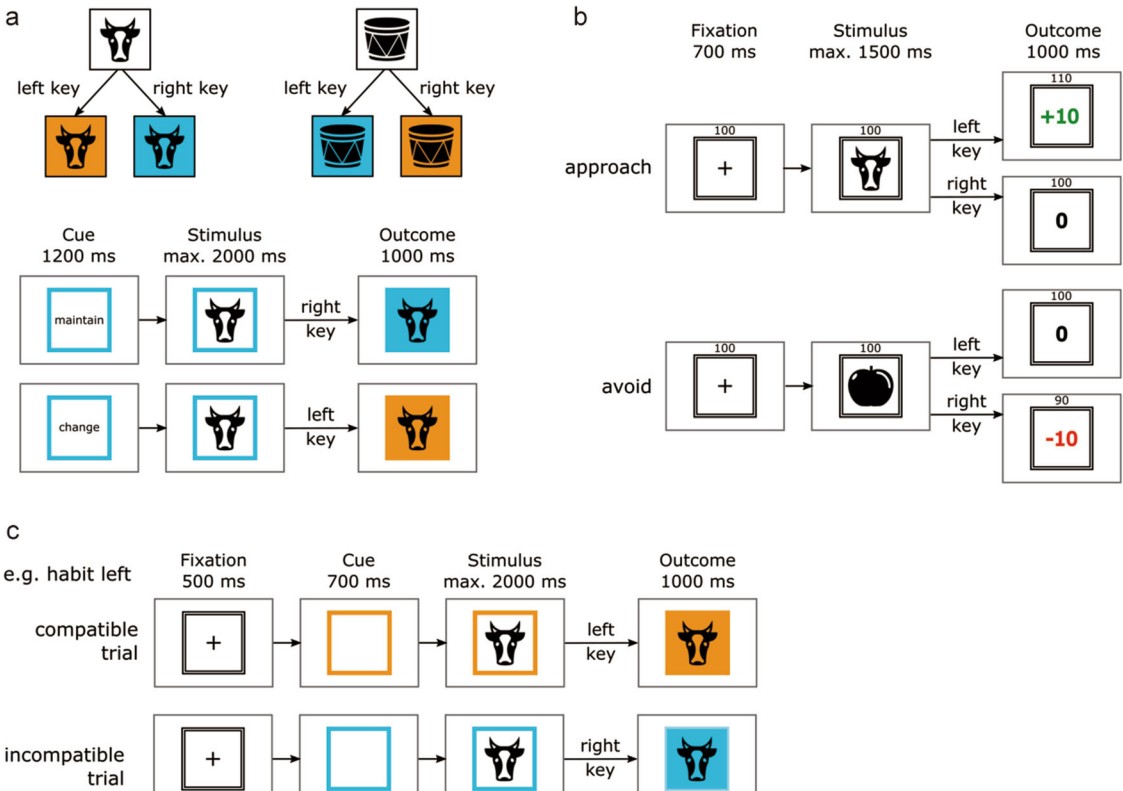

**Fig. 1 | Experimental paradigm for study 1 with three consecutive phases.** For the dynamic functional brain state analysis, only data from phase 2 were used. The purpose of phase 1 and phase 3 was to generate a behavioral index of 'habit strength'. **a** Examples of the instructed hierarchical associations between stimulus category, response, and outcome (background color) used in phase 1 and two exemplary trials from phase 1. In phase 1, goal-directed behavior was established: given a certain stimulus category (artificial or natural), participants had to execute a certain action (left or right key press) in order to change or maintain the background color (no stimulus-specific associations could be learned). **b** Exemplary trials from Phase 2. In phase 2, participants had to learn by trial-and-error stimulus-specific associations with the two response options ('habits'). In the approach condition, a correct response was indicated by monetary gain (study 1 only). In the avoidance condition, a correct response was indicated by the absence of monetary loss (study 1 and study 2). **c** Examples for compatible, incompatible trials in Phase 3. In phase 3, goal-directed responses established in phase 1 were put into competition with responses trained in phase 2 in order to probe individual habit strength.

To this end, the present study utilized network modularity analysis as a method that enables us to characterize the balance between brain network segregation and integration and reflects more directly the fundamental aspects of cortical and subcortical organization[15–19]. Indeed, there is an emerging interest in understanding how modular brain organization balances the network-level segregation and integration states associated with changing task demands[15,20–22]. On the one hand, previous research has shown that functional segregation (high modularity) is indicated by strong functional coupling within modules (i.e., communities) with little or no functional coupling across different modules. On the other hand, functional integration (low modularity) is indicated by globally strong functional coupling across different modules, including strong information flow across different networks and their mutual interconnections[22,23]. Moreover, it has been shown that dynamic transitions between states of high integration and states of high segregation are linked to different levels of attention[24], cognitive performance[16,19], and also intelligence[18]. Network modularity analysis has also been used to show that behavioral changes evolving over the course of learning (e.g., indexed by learning rate) are associated with increasingly segregated functional brain network organization[8]. But to date it remains unclear how such brain state changes might be associated not only with performance improvement during learning (i.e., learning rate) but also with actual habit strength as assessed by the post-learning goal-habit competition test. The goal of the present study was to examine both aspects concurrently, with learning rate indicating how quickly individual subjects are able to extract the new rules by trial-and-error and with habit strength as a measure of how well the newly acquired rules were

being automatized through extended practice or, in other words, how enduring the trained S-R associations are.

We proceeded in two steps. First, we tracked changes in brain network segregation and integration during the goal-habit transition. The transition between the two different brain states was quantified using dynamic sliding window functional connectivity analysis[19,25–27]. For each time window, this method estimates correlations between multiple brain regions over successive time points, which are then clustered into sets of recurring patterns, so-called dynamic connectivity states (DCS). Second, we investigated the relationship between the individual transition rate of the two different brain states (i.e., integrated vs. segregated states) and individual task performance (learning rate and habit strength) as quantified via hierarchical drift-diffusion modeling (HDDM)[28]. Importantly, to demonstrate the generalizability and robustness of all our findings, we performed all the analyses across two separate studies using similar (but not identical) experimental designs and different age groups. We hypothesized that fast learners who are characterized by reaching ceiling drift rate levels in a shorter period of time than slow learners would exhibit a faster transition from a relatively integrated brain state into a relatively segregated brain state. Moreover, we examined whether this transition would also be related to post-learning differences in habit strength.

## Results

### Learning rate

We applied HDDM[28] to investigate the changes in drift rate across the stimulus repetitions, which represents learning progress. Changes in averaged drift rate across subjects were calculated to illustrate the overall increase

**Fig. 2 | Changes in averaged drift rate across subjects during habit learning (thick lines) together with individual drift rate curves (thin lines).** Repetition in $x$ axis denotes the number of occurrences of each stimulus.

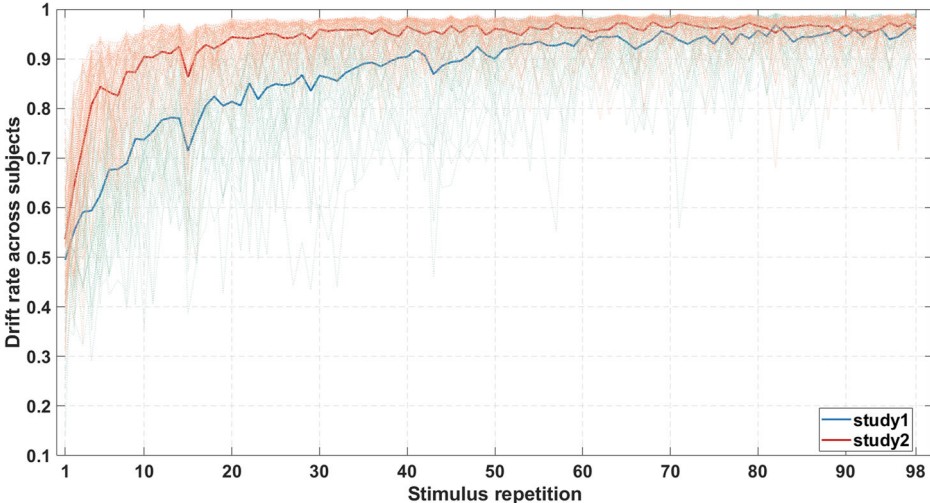

in drift rate during learning. As shown in Fig. 2, drift rate of study 2 reached ceiling level before study 1. This reflects that the learning task was easier in study 2 than in study 1. Posterior predictive checks showed good agreement between the observed data and the data generated from the model posteriors, as shown in Supplementary Fig. 1.

To obtain a single parameter estimate that represents the learning rate in each subject, the power function was fitted to each individual drift rate curve[29]. The originally estimated exponent parameter of the power function reflects the steepness of the drift rate curve, but counter-intuitively, smaller values indicate a higher learning rate. Hence, to facilitate interpretability, individual learning rate was defined as the negative exponent parameter such that a larger learning rate value represents faster learning. Accordingly, the learning rate in study 2 ($M = -0.1141$, Sd = 0.0555) was significantly higher than in study 1 ($M = -0.3769$, Sd = 0.1602), as evidenced by the Welch's $t$ test ($t = 11.3097$, $p < 0.0001$).

### Habit strength
Paired $t$ test were applied to examine the difference of individual drift rates between the compatible and incompatible conditions for study 1 and study 2 separately. The results show that the drift rates in compatible conditions were significantly higher than those in incompatible conditions in both study 1 ($M = 1.48$, Sd = 0.53 and $M = 1.35$, Sd = 0.46 for compatible and incompatible conditions, $t (49) = 3.2864$, $p = 0.0019$) and study 2 ($M = 1.35$, Sd = 0.42 and $M = 1.24$, Sd = 0.44 for compatible and incompatible conditions, $t (92) = 3.1903$, $p = 0.0019$) indicating a behavioral impact of the habit acquired in phase 2.

### Whole-brain modularity analysis
To assess the dynamics of integrated and segregated brain states during goal-habit transition in a time-resolved manner, k-means clustering was applied to the 20 (as well as 10, see Supplementary Fig. 3) tapered windowed FC matrices. As shown in Fig. 3a, each matrix represents the centroid of a cluster and putatively reflects a connectivity state stably present within the data. Descriptively, within-network connectivity strength in state 1 appeared weaker than in state 2, such as in CON and default-mode network (DMN), while the anticorrelation between different functional brain networks in state 2 appeared much stronger than in state 1, such as between SMN and DMN or between CON and DMN. More specifically, system segregation was higher in state 2 when compared with state 1 in both data sets (Fig. 3b). Thus, brain state 1 was associated with the more integrated state while brain state 2 was associated with the more segregated state. Once we had determined the integrated and segregated brain states, we then examined their prevalence as a function of time. For both data sets, the prevalence of the segregated state increased during learning, necessarily

paralleled by a decreasing prevalence of the integrated state. Since only two states were considered here, the sum of the prevalence of the segregated and integrated brain states always equals 1 (Fig. 3c). The primary purpose of the k-means clustering analysis is to identify the prevailing brain states elicited during our task. Thereby we aimed to establish that our task is dominated by two antagonistic states reflecting a more integrated vs. a more segregated brain state akin to the general pattern described before for other cognitive tasks[19]. Moreover, we aimed to establish that the prevalence of these two brain states is systematically changing across learning. Following this initial descriptive analysis, we performed a more rigorous statistical analysis based on the modularity-Q value[30,31]. This allowed us to properly quantify statistically how functional segregation evolves across learning.

As expected, and also consistent with the prevalence results, the modularity-Q value, which represents the degree of segregation, increased significantly across learning as evidenced by one sample $t$ tests on the beta coefficients of a linear regression, both in study 1 ($t (49) = 7.3624$, $p < 0.001$) and in study 2 ($t (92) = 5.3465$, $p < 0.001$). Across studies, the modularity-Q value increase in study 1 was numerically stronger than in study 2 ($t = 1.8583$, $p = 0.0656$). As depicted in Fig. 4a, a lower slope of Q does not reflect lower overall or final segregation, but rather a very early increase in segregation. In fact, the study with the overall higher learning rate (study 2) exhibited a higher modularity-Q value already in the earlier phase of learning which was hence accompanied by a smaller slope of the modularity-Q value as compared to study 1. In other words, a very steep learning curve (as in study 2) was associated with less additional increase in brain network segregation across training. As reported in the next section, a converging pattern was also revealed when analyzing the relationship between the Q value time course and individual learning rate within each study.

### Relationship between the dynamics of network segregation and behavior
Pearson correlation coefficients were calculated between the change in network segregation (slope of the modularity-Q value) and learning rate in phase 2 and habit strength in phase 3. We found a significant negative correlation between the learning rate and slope of the modularity-Q value in both study 1 ($r = -0.3318$, $p = 0.0174$) and study 2 ($r = -0.2478$, $p = 0.016$) (Fig. 4b, c), which again indicated that the subjects with a higher learning rate were the subjects who were faster in transition from a more integrated into a more segregated brain state, therefore, showing less increase in segregation across training.

To provide a more easily interpretable representation of this relationship, we performed a complementary analysis that compared the time courses of the modularity-Q value for different groups of subjects defined by

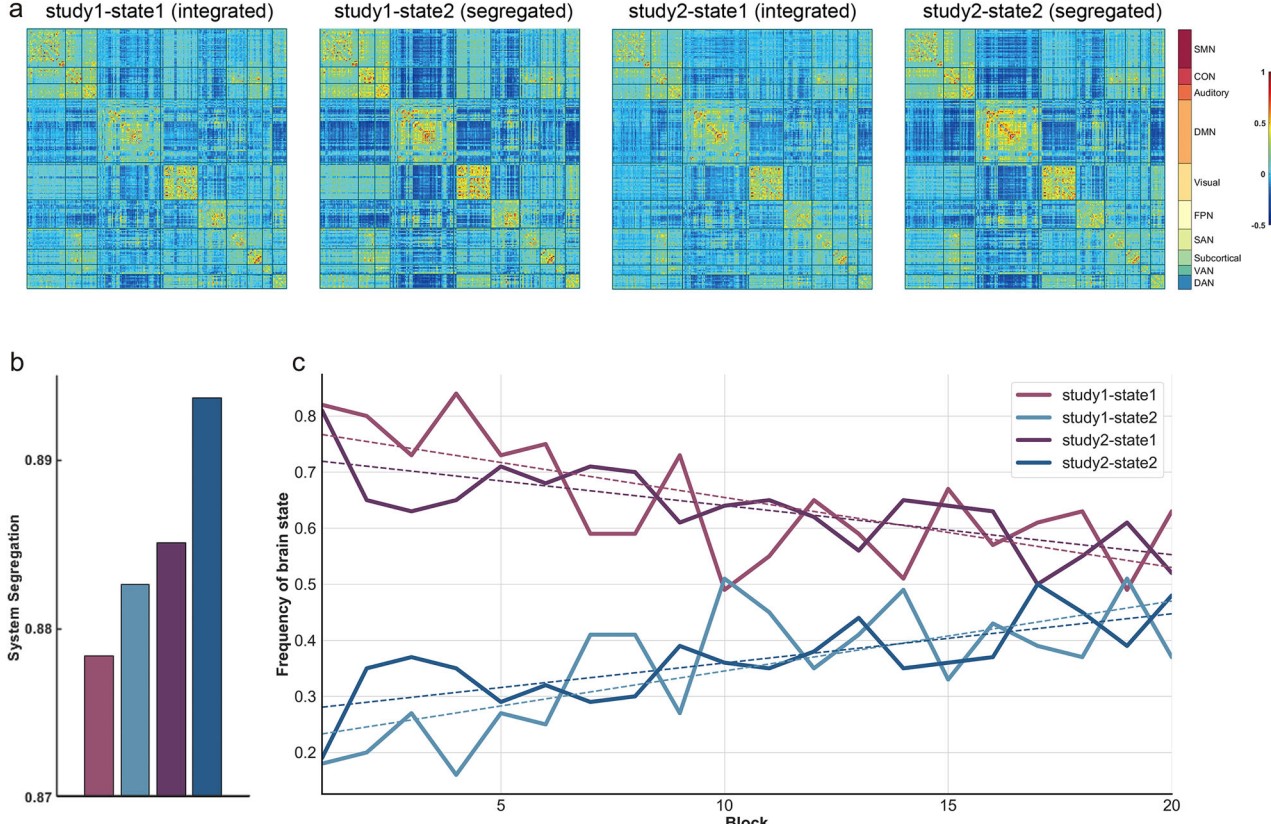

**Fig. 3 | Dynamics of brain states during learning in phase 2. a** Cluster centroids derived from k-means analysis in study 1 and study 2 are shown as connectivity matrices; **b** whole-brain system segregation computed from cluster centroid matrices. The histogram represents study 1-state 1, study 1-state 2, study 2-state 1, and study 2-state 2 from left to right separately, and the color index is the same as in **c**; **c** changes in frequency of the occurrence of integrated (state 1) and segregated brain (state 2) states during learning in study 1 and study 2.

a median split according to individual learning rate (Fig. 4d, e). This shows, for both studies, that the group of subjects exhibiting a higher individual learning rate exhibited a smaller (additional) increase in segregation (i.e., a smaller slope of the modularity-Q value) across learning since they achieved a higher modularity-Q value and higher performance levels already in the earlier learning phase. As suggested by visual inspection of Fig. 4d, e, network segregation seems to happen earlier for quicker learners (and for the easier task, see Fig. 4a). To confirm this statistically, we averaged 'early modularity values' from sliding windows 1–3 and correlated it with both learning rate and habit strength. There was a significant correlation between the early modularity values and learning rate in study 2 ($r = 0.2961$, $p = 0.0019$, one-tailed) and also in study 1 ($r = 0.2548$, $p = 0.0355$, one-tailed). However, and again as in the primary analysis, we did not find a significant correlation between the early modularity values and habit strength neither in study 1 ($r = 0.1192$, $p = 0.4049$) nor study 2 ($r = 0.0151$, $p = 0.8853$).

In addition, no significant correlations were observed between the slope of the modularity-Q value and habit strength in both study 1 ($r = -0.1955$, $p = 0.1737$) and study 2 ($r = 0.0288$, $p = 0.7852$). Refer to Supplementary Fig. 4 for similar results based on habit strength defined by the RT-based compatibility effect (higher compatibility effect indicated stronger habit strength).

#### Individual functional brain networks

While the modularity analysis described above demonstrated that the segregation of the whole-brain system was gradually increasing across learning, we also observed that the degree of segregation and integration varied between different functional brain networks. As shown in Fig. 5, the DMN exhibited an increasing segregation accompanied by a decreasing integration, which is consistent with the result of the whole-brain system

modularity analysis. This was evidenced by a positive value of the slope of the module-degree Z (MDZ) score (*y* axis) paralleled by a negative value of the slope of the participation coefficient (*x* axis). By contrast, different from most other functional brain networks, the fronto-parietal network (FPN) exhibited a decreasing segregation accompanied by a stable integration across learning. This was evidenced by the negative value of the slope of MDZ score (*y* axis) while the slope of the participation coefficient (*x* axis) remained close to zero. Statistically, one sample *t* tests conducted on each individual functional brain network with FWE-correction demonstrated that the decreased segregation of the FPN, the increased segregation of the DMN as well as the decreased integration of the DMN were significantly different from zero in both studies (Supplementary Table 1). In addition, we also observed a decreased integration of the salience network (SN) in both studies, and a decreased segregation of the auditory network and ventral attention network but only in study 1 (Supplementary Table 1). We finally correlated the individual learning rate as well as the individual habit strength score with the integration and segregation coefficients of those individual brain networks that were identified to exhibit significantly changed coefficients across learning. Importantly, we found a significant negative correlation between the increased segregation of DMN and the learning rate in both study 1 ($r = -0.2959$, $p = 0.0390$, uncorrected) and study 2 ($r = -0.2234$, $p = 0.0314$, uncorrected). No significant correlations were found regarding habit strength (all $p > 0.633$).

#### Discussion

In the present paper, we characterized how modular brain organization rebalances network-level segregation and integration during the transition from more goal-directed to more habitual behavior. We found that in the initial learning phase, which is supposed to be governed by goal-directed control, the whole-brain network was more integrated with globally strong

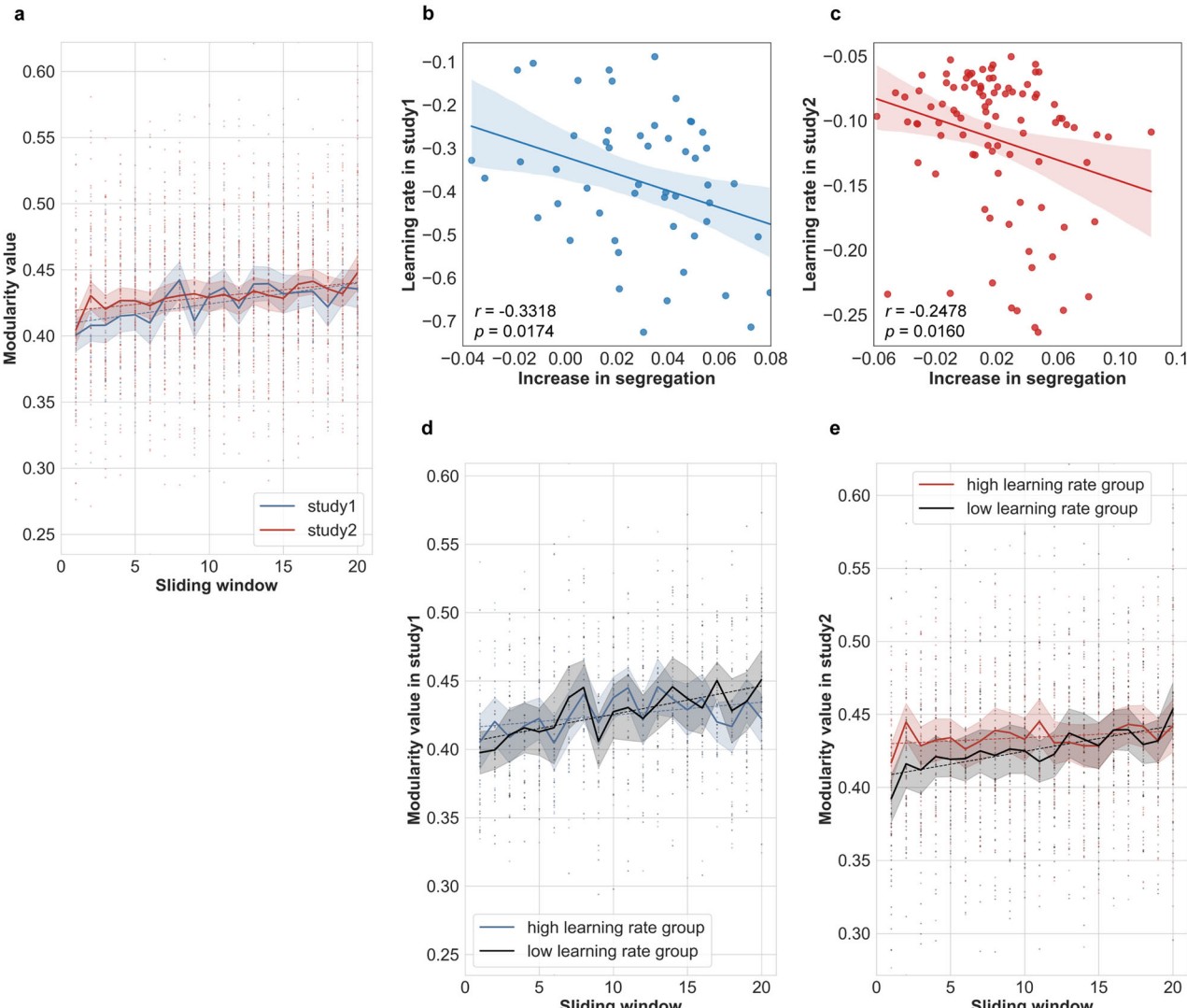

**Fig. 4 | Correlation between the dynamics of segregated network state and behavior. a** Changes in mean modularity-Q value across all subjects in each single study; correlation between the learning rate in phase 2 and the slope of modularity-Q value in study 1 (**b**) and study 2 (**c**); changes in mean modularity-Q value for high learning rate and low learning rate subgroups for study 1 (**d**) and study 2 (**e**). The shaded area represents the 95% confidence interval, and each individual data point represents one single subject. To facilitate interpretability, the individual learning rate was defined as the negative exponent parameter of the fitted power function such that a larger learning rate value represents faster learning.

functional coupling across different modules. However, across learning, the whole-brain network became more segregated, as reflected by an increased modularity-Q value. Furthermore, the transitional speed from more integrated to more segregated brain states was correlated with the individual behavioral learning rate but not with the habit strength. This suggests that brain modularity dynamics, as assessed here, are associated with performance improvement speed early during trial-and-error learning rather than with continuously evolving training-related habit strengthening as assessed by the goal-habit competition test. The primary aim of the current paper is not to identify potential neural differences between Study 1 and Study 2. Instead, we aimed to identify brain state transitions underlying S-R learning that generalize across studies. That is common learning-related brain state changes that are independent of study-related differences in task difficulty, gender distribution, and age range.

## Whole-brain modularity changes

In the field of human learning, the present results resemble previous findings from studies investigating learning either on a much slower timescale[8] or on a much faster timescale[7]. Both studies reported increasingly segregated brain networks over the course of learning and significant relationships

between network segregation and learning rate. The key difference between those previous studies and the present study relates to the type of learning. Both previous studies examined 'task automatization' of fully known task rules involving either long-term motor skill consolidation related to repeated implementation of the same motor sequence[8] or short-term consolidation of newly instructed stimulus-response rules[7]. Hence, in both studies, learning progress was behaviorally reflected by increasingly shorter response times while accuracy levels were close to optimum already from the outset and increasing network segregation was associated with response time-related learning rate. By contrast, in the present study, novel task rules had to be established in the first place via trial-and-error learning. In turn, learning progress was reflected by the increasing drift rate, which is more closely related to the traditional measure of performance accuracy rather than response times[32,33], and this was associated with increasing network segregation (please refer to Supplementary Fig. 5 for the correlation results derived from performance accuracy). Moreover, different from previous studies, the present study allowed us to compare functional network changes associated with trial-and-error learning under different task difficulties (8 S-R rules in study (1) vs. 4 S-R rules in study (2). The crucial observation was that despite very different mean learning rates in the two studies (higher

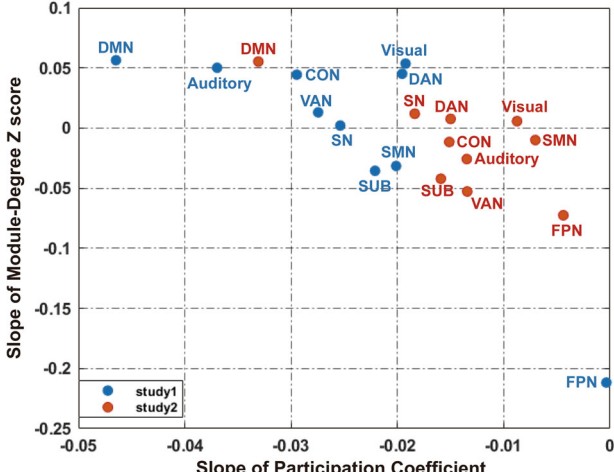

**Fig. 5 | Functional cartography of individual functional brain networks.** Each brain network is represented in a position defined by its averaged slope value of segregation and integration coefficients across subjects. CON cingulo-opercular network, SMN sensorimotor network, DAN dorsal attention network, SUB sub-cortical network, VAN ventral attention network, SN salience network, FPN fronto-parietal network, DMN default-mode network.

learning rate was associated with the easier learning task), we still found that increasing functional network segregation was commonly associated with inter-individual differences in learning rate. More specifically, in both our studies, the transitional speed from a more integrated to a more segregated network state was related to the individual learning rate such that individuals exhibiting a faster brain state transition (the individuals who reached a more strongly segregated network state earlier, and hence showed less additional increase in segregation across training) were learning faster than individuals exhibiting a slower brain state transition.

On a more general level, our current findings are consistent with previous studies which suggested that network segregation was lowest (while integration was highest) during high-demanding tasks, such as an n-back task, when compared with less demanding tasks, such as a simple motor task or resting state[15,16,19–22,34]. Results like this are in line with the Global Workspace Theory by demonstrating that less demanding, highly automated/habitual tasks can be performed within segregated modules, while more challenging controlled/goal-directed tasks require the integration between multiple modules[16,23,35].

Together, it seems that our current finding of increasing learning-related network segregation reflects a general phenomenon not only common to various types and timescales of learning but also appears to be more prevalent in less demanding task conditions. However, it should be noted that despite the common finding of increased network segregation with less demanding task conditions across a variety of different experimental paradigms, future studies need to examine potential differences in the more fine-grained structure of these connectivity changes that might potentially reveal differences depending on paradigm-specific task affordances. This can only be achieved by directly statistically comparing network changes across paradigms on the level of individual functional brain networks.

## Network-specific modularity changes

**Default-mode network.** Regarding the present study, when considering different functional brain networks individually, the DMN emerges most prominently in our dynamic network analysis. Traditionally considered a 'task-negative' functional network, the DMN normally exhibits high activity during internally oriented processing, such as mind wandering[36–39], and was anti-correlated with the activity of systems that engage in demanding cognitive tasks, such as the FPN and dorsal attention systems[16,40–43]. Previous studies have shown that the modularity of the DMN increased across working memory training[16,34]. Consistent

with that, in the present study, the modularity of the DMN increased significantly across learning, with stronger positive intra-network connectivity paralleled by stronger negative inter-network functional connectivity (Fig. 3a).

**Fronto-parietal network.** In contrast to the DMN findings that resembled whole-brain network connectivity changes, the FPN showed a different pattern. The FPN, which is considered to comprise flexible hub regions involved in high-level cognitive control[44–46], demonstrated decreasing modularity across learning in the present study. These results are consistent with previous studies, which demonstrated a positive correlation between FPN activation and S-R rule difficulty[47] as well as between the within-network functional connectivity strength of the FPN and working memory load[48]. In Repovs et al., the within-module functional connectivity was lower for lower working memory load/task difficulty, which might suggest that simpler tasks require less coordination and communication within the FPN. Applied to the present study results, this implies that the learning-related decrease in FPN segregation reflects the reduced requirement for communication within the FPN due to reduced cognitive control demands when the task becomes more automatized. All this evidence suggests that the more cognitively demanding the task, the stronger the activation of FPN as well as the within-FPN functional connectivity. Moreover, previous studies focusing on the local activity level have found a decreasing activation within the FPN during instruction-based learning and other types of learning[6,49,50]. This might suggest that the high-level computations of correct responses derived from FPN were only necessary at the early learning phase[7], but with increasing practice, high-level control of abstract S-R representations and general guidance by the FPN was needed to a lesser extent[44,51,52].

In summary, the current study provides important new insights into how inter-individual differences in trial-and-error learning are related to brain state dynamics. Replicated across two independent data sets, we demonstrated a transition from a more integrated brain state to a more segregated brain state during the goal-habit transition. Crucially, the faster this transition proceeded, the quicker the learning of novel stimulus-response associations proceeded, as evidenced by the significant correlation between the changes in brain modularity and learning rate. In contrast, we did not observe a relation between changes in brain modularity and habit strength. We also failed to find a significant correlation between learning rate and habit strength ($r = 0.0149$, $p = 0.9172$ in study 1 and $r = 0.054$, $p = 0.6049$ in study 2). Together with the significant correlation results between early modularity values and learning rate, this again suggests that learning rate indicates how quickly individual subjects are able to extract the new rules by trial-and-error. This contrasts with the absence of a significant correlation between early modularity values and habit strength as a measure of how well the newly acquired rules were being automatized through extended practice or, in other words, how enduring the trained S-R associations are. This exclusive link between brain network modularity dynamics and learning speed nicely demonstrates that profound changes in global network organization can occur rapidly already during the initial phase of rule extraction. In contrast and maybe counter-intuitively, the arguably more profound and more enduring impact of training on behavior as indexed by habit strength does not seem to leave a measurable trace on global network segregation. Importantly, this does, of course, not exclude the possibility that other neural indices might be associated with behavioral indices of habit formation. For example, previous research found that training-related brain activity changes in angular gyrus[13] and head of caudate[53] were associated with habit strength. Connectivity-wise, from our previous research, we found that the functional connectivity changes involving the sensorimotor and the CON contributed most prominently to habit strength prediction[14]. In addition, Mill et al. found that the geometry of neural representations changes, originating in the subcortex (hippocampus and cerebellum) and slowly spreading to cortex to support a transition from novice to practiced performance[54]. Moreover, by

applying representational similarity analysis, Tambini et al. found that the specific hippocampal representations emerge early, followed by both specific and schematic representations at a gradient of timescales across hippocampal-cortical networks during the longitudinal memory representation task[55].

We believe there are good reasons to argue that increased segregation observed in our study mainly reflects learning-related changes in task difficulty/automatization rather than more unspecific effects merely related to the passage of time, as would be the case during resting state studies. Specifically, brain state transitions in resting state studies seem to follow different patterns than we observed in the present study. One previous study found that the prevalence of segregated vs. integrated states during resting was fluctuating over time but there was no systematic trend in one or the other direction like in our study[19]. Moreover, another study found that segregation actually decreased across resting state scanning[25]. This is the opposite direction than in the present study and in other learning studies[7,8]. Moreover, a resting state study that involved sleep deprivation demonstrated that different arousal states were indeed associated with different brain states[56]. However, this association was opposite to what would be expected if our own present results were driven by hypothetical changes in arousal being confounded with progress in learning (i.e., high arousal in the beginning associated with a more integrated state vs. low arousal in the end associated with a more segregated state). Specifically, in Wang et al. (2015), the low arousal state was associated with a more 'integrated' brain state with reduced within-network functional connectivity of default mode and dorsal/ventral attention networks, as well as reduced anticorrelation between these networks. In contrast, the high arousal state was associated with the more 'segregated' brain state with more decoupling between the DMNs and higher-order cognitive networks that included dorsal/ventral attention networks[56]. To conclude, in light of previously published resting state studies[19,25,56] as well as previous studies on learning[7,8], our present observation of increased network segregation across learning seems most likely to be related to progress in learning, not least suggested by the significant association with learning rate.

There is still one limitation we should bear in mind when computing the functional connectivity, that is, the strategy for regressing out the average task-related activity. Cole et al.[57] found that finite impulse response (FIR) basis functions might be a better choice than others for removing the potential influence of mean task-related co-activations from estimates of connectivity. Further studies might consider using FIR or other more flexible basis functions like Fourier sets as the primary method for regressing the mean task activity when computing the task-based functional connectivity.

More generally speaking, our present findings are in line with the notion that increasing network segregation might be a common consequence of more efficient task performance not only in feedback-driven learning but also in different types of learning and task domains other than learning. An important next step for future studies will be to examine the more fine-grained structure of these large-scale network changes in order to disentangle potential domain-specific and domain-general aspects associated with differences in task efficiency.

## Methods
### Study 1
**Participants.** After excluding three subjects from further analyses due to excessive head movement (see "fMRI preprocessing" for more details), fMRI data of fifty subjects (28 females, 22 males; mean age: 23 years, range 19–31 years) were re-used from our previous study[13]. All subjects were right-handed, neurologically healthy, had normal or corrected vision, and normal color vision. The experimental protocol was approved by the Ethics Committee of the Technische Universität Dresden. All subjects gave written informed consent prior to taking part in the experiment, and they were compensated with 8€ per hour in addition to the money they gained during the experiment. All ethical regulations relevant to human research participants were followed.

**Experimental procedure.** The experimental paradigm consisted of three consecutive phases. During the first phase (goal-directed training), which was performed outside of the MRI scanner, goal-directed behavior based on hierarchical S-R-O associations was established. During the second phase (habit induction), which was performed inside the scanner, subjects were required to learn novel responses to gain monetary reward or to avoid monetary loss for a subset of the stimuli already used in the first phase. The training was continued beyond asymptotic performance levels in order to further strengthen habitual S-R associations by overtraining, during which each stimulus was repeated 98 times. Finally, during the third phase (goal-habit competition), which was also performed inside the scanner, monetary outcomes associated with habitual responses putatively established in phase 2 could no longer be obtained, and habitual responding and goal-directed actions established in phase 1 were put into competition to measure the habit strength developed in phase 2. In the current study, we utilized and analyzed fMRI data from phase 2 (goal-habit transition) and behavioral data from phase 2 (learning rate) and phase 3 (habit strength during goal-habit competition).

**Phase 1 (goal-directed training).** The purpose of phase 1 was to establish hierarchical S-(R-O) associations where the correct response upon a given stimulus category depended on the outcome to achieve. Hence, no stable associations between stimuli and responses (i.e., no S-R habits) could be learned. Ten stimuli were grouped into five "artificial" and five "natural" stimuli (natural stimuli: tree, snowflake, cow, mushroom, lungs; artificial stimuli: scissors, computer mouse, car, cupboard, ball). Responding to a stimulus from one group of stimuli (e.g., artificial) with the right key led to a blue outcome color, and responding with the left key led to an orange outcome. This R-O association was inverted for the other group of stimuli (e.g., natural), such that pressing the right key led to an orange outcome and pressing the left key led to a blue outcome. Phase 1 comprised 240 trials and each trial started with the presentation of a cue containing either the German words for "change" or "maintain". This word was framed by a colored square displaying the present outcome color that was produced in the previous trial. The subjects' task was to press the key that would either change or maintain the current outcome color for the associated category the currently displayed stimulus belonged to. For a more detailed description, see Zwosta et al.[13].

**Phase 2 (goal-habit transition).** The purpose of Phase 2 was to enable the formation of S-R habits for a subset of the stimuli already used in Phase 1. Eight of the ten stimuli (four artificial and four natural stimuli) were re-used from phase 1. In the beginning, subjects were instructed that the categories of the stimuli were now irrelevant and that they had to find out the correct key for each of the eight stimuli individually by trial and error. For both categories, each of the four stimuli belonging to one category was associated with one of the four combinations of correct responses (left or right) and outcome types (approach or avoidance). Subjects were also explicitly told that for four of the stimuli, the correct response would allow them to gain points, while for the other four stimuli, the correct response would allow them to avoid losing points. Rewards were +10 points printed in green color, punishments were −10 points printed in red, and outcomes of 0 points were printed in black. If subjects failed to execute any response during the response window, they also received an unfavorable outcome, i.e., they lost 10 points ("−10") in avoidance trials and gained zero points ("0") in approach trials. Trials in phase 2 were clustered into seven task blocks with 112 trials each (14 per stimulus). Hence, the whole phase 2 consisted of 784 trials (98 per stimulus).

**Phase 3 (goal-habit competition).** Prior to phase 3, subjects were instructed that they could no longer gain or lose any points, and thereby, the contingency between stimulus, response, and monetary outcome was removed. Hence, any tendency to continue to perform the trained

response established in phase 2 should not be motivated by aiming to gain reward or avoid loss but should be based on habitual responding instead. Phase 3 had 384 trials, and each trial started with a fixation cross, followed by a colored frame (cue), which was either one of the two outcome colors (blue and orange) previously introduced in Phase 1 or a new third color (purple) indicating free-choice trials. If the frame was blue or orange, then subjects were required to press the response that would lead to this particular outcome color for the displayed stimulus according to the R-O contingencies introduced in phase 1 (goal-directed trials). If, however, the frame was purple, then subjects could freely choose one of the two responses (free-choice trials). We were interested in two different trial categories: (1) trials for which the trained response towards the stimulus was identical to the required goal-directed response either because it had previously been rewarded or not punished (compatible condition, 96 trials in total). (2) trials for which the trained response did not match the required goal-directed response (incompatible condition, 96 trials in total). The habit strength was computed as the response time difference between the incompatible and compatible conditions.

## Study 2

The original purpose of study 2 was to investigate the potential difference in goal-habit transition between healthy controls and anorexia nervosa patients. In the current study, only the participants of the healthy control group were included.

**Participants.** After excluding two subjects from further analyses due to excessive head movement (see "fMRI preprocessing" for more details), fMRI data of ninety-four subjects (all females, mean age: 19 years, range 12–30 years) were re-used in the current study.

**Experimental procedure.** The experimental procedure in study 2 differed from the experimental procedure in study 1 such that (1) instead of the original two conditions, "approach" and "avoidance" in phase 2, we only used the avoidance condition; (2) instead of 10 stimuli, we reduced the number of stimuli to 6 (3 "artificial" and 3 "natural") and consequentially the total number of trials in each phase: phase 1 consisting of 132 trials (240 in study 1), phase 2 consisting of 392 trials (784 in study 1) and phase 3 consisting of 112 trials (384 in study 1); (3) we replaced stimuli associated with food (mushroom, cow) to prevent an adverse reaction by the anorexia nervosa patient group (which was not part of the present analysis). To this end, we used stimuli depicting a sun, a tree, and a flower for the natural category and a scissor, a car, and a ball for the artificial category.

## Statistics and reproducibility
### Behavioral analysis

**Learning rate.** A drift-diffusion model was applied to identify the conditional drift rate (v) corresponding to the rate of evidence accumulation for each round of stimulus repetition across learning for each subject. The faster the drift rate increases, the quicker learning proceeds. Compared with the mere analysis of accuracy or reaction time, the drift rate model is based on the premise that reaction time and response output can be decomposed into parameters reflecting the latent cognitive processes driving task performance[58]. This model provides a parsimonious account of complex behavioral phenomena, including response latency distributions[59] as well as speed-accuracy trade-offs[60].

Consistent with previous research on deterministic learning procedures similar to ours[61,62], the current study analyzed the behavioral measures with HDDM[28]. This choice was made mainly based on the reason that HDDM assumes that participants are random samples drawn from group-level distributions, and uses Bayesian statistical methods to simultaneously estimate parameter distributions at both the group level and the individual-participant level[28,63]. Therefore, compared with alternative diffusion fitting routines (e.g., fast-dm, EZ-dm, or DMAT), HDDM optimizes the trade-off between within and between subject random effects. It accomplishes this by

accounting for both within-subject variability and group-level similarities. Individual parameters in HDDM are constrained by a group-level distribution but can vary from this distribution to the extent that their data are sufficiently diagnostic[64,65].

To examine changes in key parameter estimates across learning in phase 2 of the experiment, two parameters, mean drift rate (v) and decision threshold (a), were allowed to vary across the 98 repetitions of eight (study 1) or four (study 2) different stimuli[61,62]. The non-decision time ($t_{er}$) and starting point (z) were estimated at the subject- and group levels, while variance parameters (variance in drift, $s_v$, non-decision time, $s_t$, and starting point, $s_z$) were estimated only at the group level. Data were accuracy-coded, such that the upper threshold (a) of the model corresponded to a correct choice, whereas the lower bound (0) corresponded to errors. We generated 10,000 samples from the joint posterior distribution of all model parameters by using Markov chain Monte Carlo methods[66]. The initial 1000 samples were discarded as burn-in to minimize the effect of initial values on the posterior inference (see Wiecki et al.[28] for more details of the procedure). To ensure model convergence, we inspected traces of model parameters and their autocorrelation to check that there were no drifts or large jumps, which would also suggest non-convergence. To further evaluate whether the model fits to the data, we ran posterior predictive checks by averaging 500 simulations generated from the model's posterior to confirm it could reliably reproduce patterns in the observed data[64,65,67].

Finally, a single learning rate parameter was determined for each subject based on the conditional drift rate across 98 stimulus repetitions during learning, averaged across eight (study 1) or four (study 2) different stimuli. More specifically, the learning rate was considered as the negative exponent parameter that was determined by fitting the one-term power function to the averaged conditional drift rates with the logarithm of the number of stimulus repetitions as an independent variable[8]. The fitting was achieved using a robust outlier correction in MATLAB (using the function "fit.m" in the Curve Fitting Toolbox with the option "Robust" and type "Lar")[8]. Welch's t test was applied to examine the differences in the learning rate between the two studies.

**Habit strength.** HDDM was applied again in phase 3 with the same procedure as for the learning rate of phase 2 to identify the individual drift rate corresponding to the rate of evidence accumulation for compatible and incompatible conditions. Paired t test was then applied to examine the potential drift rate difference between the two conditions. The individual compatibility effect was further considered as an indicator of habit strength, reflecting the impact of the trained habits on goal-directed behavior, and computed as the drift rate difference between the compatible and incompatible conditions. The compatibility effect of one subject was identified as an abnormal value (three standard deviations above the mean) in study 2, and this subject was hence discarded from further analysis. Our previous studies[13,14] computed the habit strength as the response time difference between the incompatible and compatible conditions. However, for consistency with the calculation of learning rate, habit strength was also based on the drift rate derived from HDDM in the current study. Please refer to the Supplementary Fig. 4 for the results based on response times.

### fMRI data analysis
**MRI scanning.** MRI data were acquired on a 3 T Siemens whole body Trio System (Erlangen, Germany) equipped with a 32-channel head coil. Ear plugs dampened scanner noise. Structural images were acquired using a T1-weighted sequence (TR = 1900 ms, TE = 2.26 ms, T1 = 900 ms, flip = 9°) with a resolution of 1 mm × 1 mm × 1 mm. Functional images were acquired using a gradient echo-planar sequence (TR = 2000 ms, TE = 30 ms, flip angle = 80° in study 1; TR = 2070 ms, TE = 25 ms, flip angle = 80° in study 2). Each volume contained 32 slices (4 mm, 20% gap) that were measured in ascending order in study 1, whereas 36 slices (3.2 mm, 20% gap) were measured in descending order in study 2.

**fMRI preprocessing.** Data preprocessing was performed with SPM12 running in Matlab 9.5. The same processing steps were applied to study 1 and study 2 which included the following steps: discarding the initial 3 volumes; slice timing correction; motion correction; co-registering the T1-weighted images to the mean functional images and segmenting it into gray matter (GM), white matter (WM) and cerebrospinal fluid (CSF); spatial normalization (3 mm resolution); nuisance regression which included the original six motion parameters, average signals in WM, CSF masks and their expansions (the first-order temporal derivative, as well as their squares and squared derivatives) as well as the whole-brain signal; spatial smoothing (6 mm FWHM).

In order to improve the test-retest reliability[68] and to reduce spurious correlations between different brain regions in task-based functional connectivity analyses, the average task-related activity was regressed out[57]. To this end, we performed the single-subject GLM analysis to obtain the residual timeseries for each subject which were then used for further functional connectivity computation. Learning trials were assigned to the correct approach or avoidance and error trials (irrespective of approach and avoidance) in Study 1 and correct avoidance and error trials in Study 2 separately. To appropriately capture BOLD activation, we used Fourier basis set regressors, including 14 different sine-wave regressors spanning 30 s, which were time-locked to the onset of the learning trials. After that, only for study 1 (no voluntary break in study 2), breaks between task blocks were also included as regressors with an additional GLM, the break-related regressors were based on the standard hemodynamic response function of SPM12 and convolved with the duration of breaks which varied considerably. With each subject-specific GLM, the high-pass filter was set to a cutoff of 128 s in SPM12, and estimated with ordinary least squares (that is, AR (1) off).

Since even small head motion can confound functional connectivity analyses, subjects with spike events, diagnosed as the frame-wise displacement over 0.2 mm in >20% of the fMRI data samples, were excluded from further analysis. Three subjects in Study 1 and two subjects in Study 2 were excluded. In addition, we also tested whether head movement artifacts were responsible for individual differences in brain state dynamics. No correlation was observed between the measures of head movement artifacts (mean relative and max absolute head displacement) and the slope of modularity-Q value from the modularity analysis mentioned below for both study 1 and study 2 (all $p$ value > 0.05).

**Dynamic functional connectivity.** The signals across all voxels within each ROI were averaged and Fisher $z$-transformed for the functional connectivity analysis. The dynamic functional connectivity across the refined 227 Power nodes[7,51,69] associated with 10 different functional networks was then calculated using 20 exponentially tapered sliding windows without overlap in study 1 and half-window overlapping in study 2. The duration of the sliding windows and overall duration are slightly different among subjects depending on their individual response speed. There were 784 and 392 learning trials in Study 1 and Study 2. The average number of scans across subjects is 1366.92 in study 1 and 655.09 in study 2, since the whole task was divided into 20 non-overlapping windows in study 1 and 20 overlapping windows with half-window step in study 2. We chose overlapping windows in study 2 to obtain the same number of windows as well as duration of windows as in study 1, considering that timeseries were only half as long as in study 1. The average duration of the sliding window is 68.35 scans in Study 1 and 65.51 scans in Study 2. To make sure that our results are independent of the arbitrary choice regarding number, overlap, and length of the sliding window, we repeated the analyses using 10 tapered sliding windows without overlap. This supplementary analysis showed similar results as the original analysis in both data sets (Supplementary Fig. 3). Please refer to Pozzi et al.[70] for the codes of exponentially tapered sliding windows analysis.

Tapering provides better suppression of spurious correlations and may reduce sensitivity to outliers and was defined by the weight vector $w_t = w_0 \, e^{(t-T)/\theta}$, $t = 1, \ldots, T$, and $w_0 = (1-e^{-1/\theta})/(1-e^{-T/\theta})$. The parameter $t$ is the $t$th time point within the sliding window, $N$ is the sliding window length, and the exponent $\theta$ controls the influence from distant time points. $\theta$ was set to a third of the window length, consistent with previous studies[26,70,71]. We then constructed a functional connectivity matrix by computing the weighted Pearson correlation between timeseries of any two nodes $x_t$ and $y_t$ for each time window, and finally Fisher $z$-transformed the resulting weighted Pearson correlation matrix for the subsequent analyses:

$$r_w = \frac{\sum_{t=1}^{T} w_t (x_t - \bar{x})(y_t - \bar{y})}{\sqrt{\sum_{t=1}^{T} w_t (x_t - \bar{x})^2} \sqrt{\sum_{t=1}^{T} w_t (y_t - \bar{y})^2}}$$

where $\bar{x} = \frac{\sum_{t=1}^{T} w_t x_t}{T}$ and $\bar{y} = \frac{\sum_{t=1}^{T} w_t y_t}{T}$.

**K-means clustering.** A k-means clustering algorithm was applied to all windowed connectivity matrices (subjects × windows) using city block distance as the similarity measure. Since the current study focused on the integrated and segregated brain state transition from goal-directed to habitual behavior, as done in previous studies[19,72], the number of states/clusters (k) was set to two, and each connectivity matrix window was assigned to one of two states. Clustering was repeated 10 times with random initialization of starting centroid locations. Here we chose a two-state k-means solution mainly for two reasons. First, previous studies have demonstrated that cognitive processes are mainly driven by two antagonistic brain states (i.e., integrated and segregated), and the pattern of brain states identified at different k values was highly similar to the pattern identified for $k = 2$[19]. Since the major purpose of the current study was to investigate the evolution of these two antagonistic brain states during feedback-driven stimulus-response (S-R) learning, we set $k = 2$ as an a-priori choice here. Second, based on our current data we confirmed for both studies that $k = 2$ was a reasonable choice compared to alternative $k > 2$ solutions. Most importantly, the $k = 2$ solution resulted in one more integrated state and one more segregated state, respectively. Moreover, the relative prevalence of both states changed with learning (Fig. 3). Please refer to Supplementary Fig. 2 for more detailed quantitative information regarding alternative clustering solutions[25,73].

The system segregation[15,74] of each resulting FC cluster centroid was computed to examine the relative strength of within-network connectivity compared to between-network connectivity of the two different brain states derived from k-means:

$$system \; segregation = 1 - \frac{\bar{z}_b}{\bar{z}_w}$$

where $\bar{z}_w$ represents the mean connectivity strength of edges between all pairs of nodes within the same network and $\bar{z}_b$ represents the mean connectivity strength of edges between all pairs of nodes that spanned two different brain networks.

Finally, the frequency of the occurrence of each brain state in each time window was computed as the proportion of a number of subjects classified in that brain state.

**Modularity-Q.** The Louvain modularity algorithm from the brain connectivity toolbox (BCT)[30] was applied to investigate the optimal modular structure within the functional connectivity matrix by optimizing a quality function Q that maximizes within-module connectivity and minimizes between-module connectivity[30,31]. The modularity-Q value is a widely used index when considering the whole brain as a globally interconnected modular system. Higher modularity values (Q), therefore, indicate stronger separation of networks. For each time window, the community assignment for each node, within which each node was assigned to its own community, was assessed 500 times, and a consensus partition was identified using a fine-tuning algorithm from the BCT (http://www.brain-connectivity-toolbox.net/), which afforded an

estimate of both the modularity values and community assignment for further analysis. All graph theoretical measures were calculated on weighted and signed connectivity matrices to avoid the use of arbitrary thresholds, overcoming limitations of information loss[18,26,30,34]. The γ parameter was set to 1[19].

**Segregation transition rate.** The Louvain modularity algorithm mentioned above was applied to all the 20 windowed connectivity matrices for each subject separately. Thereby, each subject has 20 consecutive modularity-Q values across learning. To capture the segregation transition rate of each subject during learning, a linear regression was fitted to the individual modularity-Q value with the logarithm of the number of total trials as the independent variable. Welch's $t$ test was again applied to examine the difference in the segregation transition rates between the two studies. In addition, the subjects were further divided into fast and slow learning subgroups based on the median value of learning rates for each single study separately. The mean segregation transition curves across each subgroup were then plotted to characterize the segregation transition pattern that was associated with each group.

**Correlation between network segregation dynamics and behavior**
The slope of the modularity-Q value curve across 20 consecutive windows was used as a measure of each subject's network segregation dynamics. Pearson correlations between the dynamics of network segregation and the learning rate in phase 2 and habit strength in phase 3 were calculated to examine the relationship between fluctuations in network topology and behavioral performance in goal-habit transition.

**Individual functional brain networks**
To investigate the integrational and segregational role of each individual functional brain network during learning, we applied two nodal metrics here: the participation coefficient (PC) and MDZ. The PC and the MDZ score are complementary indices that allow for more fine-grained conclusions regarding individual brain networks/modules. PC quantifies the extent to which a region connects across all modules (i.e., between-module strength or the degree of integration), while MDZ score is the z score of a node's within-module strength (i.e., within-module strength or the degree of segregation)[19,30,69]. We followed a previous suggestion to combine PC and MDZ into a two-dimensional plot or cartographic profile[19]. Both the PC and MDZ were computed using the BCT[30]. The mean PC and MDZ for each individual functional brain network node were computed separately for each time window. As described above, a linear regression was again fitted to the individual PC and MDZ value with the logarithm of the number of total trials as an independent variable for each subject. We then projected the mean slope value of PC and MDZ across subjects from each functional brain network into a 2-dimensional interaction space[19]. Finally, one sample $t$ tests were computed based on the slope value of PC and MDZ across subjects for each brain network separately.

**Reporting summary**
Further information on research design is available in the Nature Portfolio Reporting Summary linked to this article.

## Data availability
The data that support the findings of this study are available on request from the corresponding author. The source data behind the graphs in the paper can be found in Supplementary Data 1.

## Code availability
The codes that support the findings of this study are available here: https://github.com/xiaoyu-TUD/functional-brain-network-segregation.

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

## Acknowledgements
This work was supported by the German Research Foundation (SFB 940 projects A2, C3, Z2; EH 367/5-1, EH 367/7-1) and the Swiss Anorexia Nervosa Foundation.

## Author contributions
X.W. contributed in writing—original draft, formal analysis, visualization, conceptualization, methodology, software; K.Z., J.H., I.B. contributed in writing—review & editing, conceptualization, methodology; S.E. contributed in writing—review & editing, conceptualization, methodology, resources, project administration, funding acquisition; U.W., H.R. contributed in writing—review & editing, conceptualization, methodology, software, resources, supervision, project administration, funding acquisition.

## Funding

## Competing interests
The authors declare no competing interests.
