## [Peer Review File · Communications Biology]

Reviewers' comments:

Reviewer #1 (Remarks to the Author):

The current study has investigated how inter-individual differences in trial-and-error learning are related to brain state dynamics. The results suggest that the transition from a more integrated brain state to a more segregated brain state occurs during goal-habit transition. The manuscript is clearly presented and the methodological approaches seem correct. Nevertheless, I have several concerns regarding the comparison between Study 1 and 2, and some aspects regarding the separation between segregated and integrated networks that may undermine the relevance of the results.

Here is a detailed description of the major issues.

1) Neuropsychological maturation does not end in the early years of life, rather it extends until the end of the second decade or more. More precisely, executive functions and learning abilities are known to evolve during adolescence. The current study merges two datasets (Study 1 and 2) that were acquired in different populations. Study 1 was conducted on young adults, 28 females and 22 males, with mean age 23 years (range 19-31 years). Study 2 included all females with a mean age of 19 years, range 12-30 years. Study 2 therefore included a large bias towards gender and age. Comparing both the behavioural and fMRI results from these two populations is challenging and it should be discussed with care.

2) The choice of a two-state k-means seems arbitrary. The manuscript should provide evidence that a $k=2$ effectively explained most of the variance.

3) A related issue concerning the choice of $k=2$ for the k-means clustering is visible in Figure, where the changes in frequency of the occurrence of integrated (state 1) and segregated brain (state 2) states during learning in study 1 and study 2 are symmetrical with respect to 0.5. In fact, by definition, the frequency of occurrence of integrated state (state 1) is equal to 1 minus the occurrence of segregated state (state 2). I wonder whether this mutual exclusion between integrated and segregated states does add a bias in the analysis. I suggest clarifying and discussing this point in the revised version of the paper.

I hope these comments will allow the authors to ameliorate the quality of the manuscript.

Reviewer #2 (Remarks to the Author):

The present report links fMRI measures of brain network segregation to learning processes evoked during two instantiations of a flexible visual object categorization task. Whole-brain network segregation (modularity) was shown to increase as learning progressed, which was significantly associated with individual differences in the drift diffusion-model estimated behavioral learning rate, but not behavioral habit strength. Breaking down the analysis into individual networks revealed the default mode network (DMN) as most prominently paralleling the whole-brain profile, whereas the fronto-parietal network (FPN) showed the inverse pattern i.e. decreasing segregation/modularity over learning.

The authors tackle an important research question as to the network dynamics underpinning cognitive task learning, and the paper is well written overall. However, I had issues with interpreting some of the results e.g. the counterintuitive direction of the modularity-learning rate relationship, and the counterintuitive decrease in FPN segregation over learning. These and other concerns are highlighted below.

Major concerns:

1. Abstract: "Overall, our current study results highlight the utility of dynamic functional brain state analysis and align with a framework that conceptualizes brain network segregation as a *general feature of processing efficiency and generalizes beyond the domain of learning*." I don't think such a statement alluding to the "domain generality" of the studied effects is justified, given that both Study 1 and Study 2 are highly similar in task structure, and use the same visual object stimuli. Indeed, this is a limitation of the present report that should be acknowledged in the Discussion. Certainly, such statements of "domain-generality" should be toned down throughout.
2. Including a schematic of the task design would greatly help the reader in grasping specifics of the task.
3. Results Pg 10/11: "We found a significant negative correlation between the learning rate and slope of the modularity-Q...subjects with a higher learning rate were the subjects who were faster in transition from a more integrated into a more segregated brain state therefore showing less increase in segregation across training". The negative direction of this behavioral association is counterintuitive, and could do with some more unpacking. The analyses in Fig 3D/E that break down the modularity timecourses by fast and slow learners are helpful. These suggest that the key difference in network segregation between these subject groups appears *early* in learning, as increased segregation for fast learners (especially for Study 2, Fig 3E), which likely underpins the slope differences. Could the authors target this early effect more focally e.g. correlate the early modularity values (perhaps averaging them across block 1-3) with behavioral learning rate and habit strength? This might yield a more intuitive positive association with learning rate, and complement the current analyses.
- 4a. The authors should better justify why learning rate and habit strength are treated as independent processes in their analysis. Intuitively, the extent to which task rules are learned more rapidly should lead to quicker automation (possibly involving rule consolidation), and hence associate with stronger habit formation; indeed, is there a correlation between the two behavioral measures? Better articulation of how learning rate differs process-wise from habit strength in the introduction might also better convey the novel contribution of this paper versus previous work (e.g. the Basset et al., 2015, NatNeuro paper).
- 4b. In interpreting the observed lack of behavioral association between network segregation and habit strength, the authors correctly highlight the possibility that their brain measures are not sensitive enough (Pg 21): "Importantly, this does of course not exclude that other indices of brain activity or connectivity dynamics might be associated with habit-related processes 13,14,51". This important point would be strengthened by citing relevant recent work looking at multivariate/representational brain measures of learning (Mill & Cole (2023), "Neural representation dynamics reveal computational principles of cognitive task learning"; Tambini et al (2023): "Structured memory representations develop at multiple time scales in hippocampal-cortical networks"). Evaluating this work might also inform a necessary discussion as to whether neural associations with habit strength were not obtained in the present report due to looking at a relatively short window of learning. Stronger associations with habit strength would perhaps emerge from more extended task learning and consolidation.
5. I had some difficulty in interpreting the FPN learning dynamics. Pg 19: "The FPN, which is considered to comprise flexible hub regions involved in high-level cognitive control 42-44, demonstrated decreasing modularity across learning in the present study. These results are consistent with previous studies which demonstrated a positive correlation between FPN activation and S-R rule difficulty 45 as well as between the within-network functional connectivity strength of the FPN and working memory load 46." This needs to be unpacked more clearly - are the authors suggesting that the FPN is somehow tracking the decrease in task difficulty over learning? Still, this does not explain why this neural tracking would take the form of a decrease in network segregation per se (versus e.g. a first-order decrease in network activation). I think it would help to provide results for FPN (and perhaps DMN, which the authors also focus on) in the same format as Figure 3D/E, and (per my point 3 above) also interrogate whether the FPN modularity slope effects are driven by an early increase in modularity for slow learners (who experience the task as more difficult than fast learners). This would provide some support for the authors' hypothesized tracking of task difficulty by the FPN's segregation.

Minor concerns:

1. Intro pg 3: "With increasing practice, however, behavior is assumed to become less and less governed by outcome anticipation and instead to become more and more controlled directly by stimulus-response (S-R) associations". The use of "control" here is confusing as it connotes "cognitive control", which could be argued to decrease as learning progresses towards automaticity. Please consider rewording.
2. To help the reader, Figure 2 should include labels linking state 1 to network integration and state 2 to network segregation.
The habit strength results on pg 11 should mention how to interpret the sign of habit strength i.e. positive values = stronger habit formation.
3. Results pg 13: please include some justification for transitioning from using modularity alone for the preceding behavioral analysis to using modularity and participation coefficient for these individual network results.

Reviewer #3 (Remarks to the Author):

The authors examine the relationship between large scale brain network dynamics and learning. They report that a faster learning rate is associated with a faster shift to network segregation relative to network integration. This relationship was evident with learning rate, but not habit strength. The authors identify similar results in two different datasets and clearly describe their scientific process from the descriptions of data analysis to their interpretation of the results. The manuscript is well written. I have a few suggestions that may improve the paper.

I could not find a description of the duration of the sliding windows used or the overall duration of the task phase.

The authors cite Cole et al., 2019 for the approach of removing mean task activation and calculating functional connectivity from the residuals of said model. However, Cole et al. also found that FIR is more effective at removing the possible inflating influence of task co-activation from estimates of connectivity. The authors here use a canonical HRF in their task model. The authors should consider using a FIR approach or acknowledging that their estimates of connectivity may be inflated to some degree.

One major question I was left with after reading the manuscript was: Are the network shifts to segregation actually related to learning, or is there another possible explanation? The authors correlate these shifts in network topology to individual differences in learning which provides some support for the authors' interpretation. However, I wonder if the same shifts in network topology would be observed during rest or during the performance of an unrelated task. Is it possible that network segregation increases as scans get longer in general? Could the changes in network topology be related to participants spending less "time on task" as the experiment progresses? Or could the start of a scan be related to increased arousal/vigilance, which wanes over time much like the shift from integration to segregation? The manuscript would be stronger if the authors could address some of these questions and provide additional support for the conclusion that network topology reflects learning rather than other possible explanations.

Reviewers' comments:

Reviewer #1 (Remarks to the Author):

The current study has investigated how inter-individual differences in trial-and-error learning are related to brain state dynamics. The results suggest that the transition from a more integrated brain state to a more segregated brain state occurs during goal-habit transition. The manuscript is clearly presented and the methodological approaches seem correct. Nevertheless, I have several concerns regarding the comparison between Study 1 and 2, and some aspects regarding the separation between segregated and integrated networks that may undermine the relevance of the results.

Here is a detailed description of the major issues.

1) Neuropsychological maturation does not end in the early years of life, rather it extends until the end of the second decade or more. More precisely, executive functions and learning abilities are known to evolve during adolescence. The current study merges two datasets (Study 1 and 2) that were acquired in different populations. Study 1 was conducted on young adults, 28 females and 22 males, with mean age 23 years (range 19-31 years). Study 2 included all females with a mean age of 19 years, range 12-30 years. Study 2 therefore included a large bias towards gender and age. Comparing both the behavioural and fMRI results from these two populations is challenging and it should be discussed with care.

We thank the reviewer for raising this important question.

We actually see the different characteristics of the two samples employed in the two studies as a strength of the paper. The primary aim of the current paper is *not* to identify potential neural differences between study 1 and study 2. Instead, we aimed to identify brain state transitions underlying S-R learning that generalize across studies. That is, common learning-related brain state changes that are independent of study-related differences in task difficulty (as evidenced by the **Figure 2**, study 1 is more difficult than study 2), gender distribution, and age range.

We had already tried to make this point in the original submission (page 18, lines 292-296). To further clarify this, we have now added this explanation: 'The primary aim of the current paper is *not* to identify potential neural differences between study 1 and study 2. Instead, we aimed to identify brain state transitions underlying S-R learning that generalize across studies. That is, common learning-related brain state changes that are independent of study-related differences in task difficulty, gender distribution, and age range.'

2) The choice of a two-state k-means seems arbitrary. The manuscript should provide

evidence that a $k=2$ effectively explained most of the variance.

We appreciate the reviewer for bringing up this insightful question.

In general, the k-means analysis was meant to be a first step mainly for descriptive purposes. We wanted to establish two basic properties of learning-related functional connectivity changes as a precondition for further, more detailed analyses. First, for consistency with previous study results (e.g. Shine et al. 2016), we wanted to show also for the present study that functional connectivity is dominated by two antagonistic brain states that are characterized by clear differences regarding segregation/ integration. Second, we wanted to make sure that these two states are associated with learning (i.e. change across learning). Following up on the descriptive confirmation that the more segregated brain state is becoming more frequent across learning while the more integrated brain state is becoming less frequent, we then conducted the more rigorous statistical analysis based on the modularity Q values.

This general point is now made more explicit on page 10, line 173-181: ‘The primary purpose of the k-means clustering analysis is to identify the prevailing brain states elicited during our task. Thereby we aimed to establish that our task is dominated by two antagonistic states reflecting a more integrated vs. a more segregated brain state akin to the general pattern described before for other cognitive tasks (Shine et al. 2016). Moreover, we aimed to establish that the prevalence of these two brain states is systematically changing across learning. Following this initial descriptive analysis, we performed a more rigorous statistical analysis based on the modularity Q value (Rubinov et al., 2010; Newman, 2004). This allowed us to properly quantify statistically how functional segregation evolves across learning.’

We chose a two-state k-means solution based on previous study results demonstrating that interindividual differences in cognitive processes are mainly associated with the two antagonistic (integrated vs. segregated) brain states (Shine et al., 2016). They showed that brain state patterns identified for $k>2$ did not add much information on top of the 2-state solution. In other words, the integrated and segregated brain states seem to be the dominant brain states in shaping the cognitive processes under investigation. Accordingly, we wanted to investigate the dynamics of exactly these two brain states (integrated and segregated) during feedback-driven stimulus-response learning and therefore set $k=2$ as an a-priori choice.

Our $k=2$ choice proved to be a reasonable one when compared to alternative $k>2$ solutions considering a variety of standard criteria provided by the dynamic BC toolbox (Sihouette, CalinskiHarabasz, and DaviesBouldin) (Liao et al., 2011, Allen et al., 2014). The three standard criteria from the dynamic BC toolbox suggested $k = 3, 2,$ and 6 in *study 1* and $k = 2, 2,$ and 5 in *study 2* (**Supplementary Figure 2**). Following the guidelines of the dynamic BC toolbox, the optimal k-value is given by the mean value of these three criteria (Liao et al., 2011). Therefore, the best k values are 4 in *study 1* and 3 in *study 2* (but note that the DaviesBouldin criterion seems to be a bit of

an outlier compared to the other two criteria in both studies. Using the mean might not be the best metric here. Using the median instead would result in k values of 3 and 2 respectively).

As the reviewer suggested, we also considered the standard elbow plots to assess explained variance. These indicate that 3 clusters explained most of the variance in both studies. Hence, according to all these criteria, $k > 2$ solutions would be the optimal choice in both studies. However, we nevertheless decided to stick with the more parsimonious $k = 2$ solution as it turned out that only two states (the most dominant ones) reflected the properties we were really interested in. As can be seen from the **Supplementary Figure 2**, the fourth and third brain states in *study 1* are rather ‘steady’ with little meaningful change across learning. Moreover, the third and fourth states are only weakly present in all time windows (5.88% for state 4 and 14.71 % for state 3) compared to the other two brain states (52.16% for state 1 and 27.25 % for state 2). In *study 2*, the third state strongly resembled the second state and was therefore considered to be rather redundant. Indeed, the system segregation values (0.867 in state 1, 0.913 in state 2, and 0.892 in state 3) also suggest that the first two states most clearly differentiate between the level of segregation whereas state 3 lies in-between. All things considered, we reasoned that the two most dominant states 1 and 2 carry the most relevant information with respect to our specific purposes. Hence, to keep focus on the most relevant aspects of the FC data, we decided to present the $k = 2$ solution in the main part of the paper. To make this choice more transparent we have now added the additional information regarding different clustering solutions in the **supplementary material**.

In the main text we added the following information on page 37, line 691-703: ‘Here we chose a two-state k-means solution mainly for two reasons. First, previous studies have demonstrated that cognitive processes are mainly driven by two antagonistic brain states (i.e., integrated and segregated) and the pattern of brain states identified at different k values was highly similar to the pattern identified for $k = 2$ (Shine et al., 2016). Since the major purpose of the current study was to investigate the evolution of these two antagonistic brain states during feedback-driven stimulus-response (S-R) learning, we set $k = 2$ as an a-priori choice here. Second, based on our current data we confirmed for both studies that $k = 2$ was a reasonable choice compared to alternative $k > 2$ solutions. Most importantly, the $k = 2$ solution resulted in one more integrated state and one more segregated state, respectively. Moreover, the relative prevalence of both states changed with learning (**Figure 3**). Please refer to the **Supplementary Figure S2** for more detailed quantitative information regarding alternative clustering solutions (Liao et al., 2011, Allen et al., 2014).’

Supplementary Figure S2. The standard elbow results and three different methods determining the optimal number of clusters in k-means clustering for study 1 (a) and study 2 (b). The optimal number from the three standard criteria (Silhouette, CalinskiHarabasz, and DaviesBouldin) from the dynamic BC toolbox was 3, 2, and 6 in study 1 and 2, 2, and 5 in study 2 as highlighted by red circles. Following the guidelines of the dynamic BC toolbox, the optimal k-value is given by the mean value of these three criteria. Therefore, the best k values are 4 in study 1 and 3 in study 2. Please note that the results derived from the DaviesBouldin strongly deviates from the other two methods. Using the median instead would result in optimal k values of 3 and 2, respectively. Cluster centroids derived from k-means analysis in study 1 (c) and study 2 (d) are shown as connectivity matrices. The fourth and third brain states in study 1 are rather ‘steady’ with little meaningful change across learning. Moreover, the third and fourth states are only weakly present in all time windows (5.88% for state 4 and 14.71% for state 3) compared to the other two brain states (52.16% for state 1 and 27.25% for state 2). In study 2, the third state strongly resembled the second state and was therefore considered to be rather redundant. The evolution of different brain states is depicted in terms of frequency of occurrence of each brain state across consecutive learning blocks in study 1 (e) and study 2 (f).

3) A related issue concerning the choice of $k=2$ for the k-means clustering is visible in Figure, where the changes in frequency of the occurrence of integrated (state 1) and segregated brain (state 2) states during learning in study 1 and study 2 are symmetrical with respect to 0.5. In fact, by definition, the frequency of occurrence of integrated state (state 1) is equal to 1 minus the occurrence of segregated state (state 2). I wonder whether this mutual exclusion between integrated and segregated states does add a bias in the analysis. I suggest clarifying and discussing this point in the revised version of the paper.

We thank the reviewer for making this helpful comment.

Indeed, when setting $k = 2$ the frequency of occurrence of the integrated state (state

1) logically has to be equal to 1 minus the occurrence of the segregated state (state 2). As mentioned in our reply to preceding point 2, even if more states are allowed the two prevailing brain states are still the integrated and the segregated states already identified with $k=2$. Please see our reply to point 2 for why we preferred to use the 2-state solution over $k>2$ solutions.

We added the following information on page 10, line 170-173: 'For both data sets, the prevalence of the segregated state increased during learning, necessarily paralleled by a decreasing prevalence of the integrated state. Since only two states were considered here, the sum of the prevalence of the segregated and integrated brain states always equals 1.'

Reviewer #2 (Remarks to the Author):

The present report links fMRI measures of brain network segregation to learning processes evoked during two instantiations of a flexible visual object categorization task. Whole-brain network segregation (modularity) was shown to increase as learning progressed, which was significantly associated with individual differences in the drift diffusion-model estimated behavioral learning rate, but not behavioral habit strength. Breaking down the analysis into individual networks revealed the default mode network (DMN) as most prominently paralleling the whole-brain profile, whereas the fronto-parietal network (FPN) showed the inverse pattern i.e. decreasing segregation/modularity over learning. The authors tackle an important research question as to the network dynamics underpinning cognitive task learning, and the paper is well written overall. However, I had issues with interpreting some of the results e.g. the counterintuitive direction of the modularity-learning rate relationship, and the counterintuitive decrease in FPN segregation over learning. These and other concerns are highlighted below.

Major concerns:

1. Abstract: "Overall, our current study results highlight the utility of dynamic functional brain state analysis and align with a framework that conceptualizes brain network segregation as a *general feature of processing efficiency and generalizes beyond the domain of learning*." I don't think such a statement alluding to the "domain generality" of the studied effects is justified, given that both Study 1 and Study 2 are highly similar in task structure, and use the same visual object stimuli. Indeed, this is a limitation of the present report that should be acknowledged in the Discussion. Certainly, such statements of "domain-generality" should be toned down throughout.

We thank the reviewer for raising this point.

We think this might be a misunderstanding and we apologize for the confusion. Actually, "domain generality" here refers to the fact that our present findings are consistent with similar results from previous studies investigating rather different types of learning (Bassett, et al., 2015; Mohr, et al., 2016) and even very different paradigms outside the learning domain (e.g. Shine et al., 2016, Vatansever et al., 2015). It was *not* meant to characterize the consistency of results for the two experiments in the current study. Regarding the (dis-)similarity of the two experiments we just wanted to highlight that the results found for experiment 1 generalize to experiment 2 despite different sample properties and despite differences in task difficulty. But in this context, we clearly would not speak of 'domain generality'.

To clarify this point, we have modified the final sentence in the Abstract (page 2, line 26-31): 'Overall, our current study results highlight the utility of dynamic

functional brain state analysis. From a broader perspective taking into account previous study results, our findings align with a framework that conceptualizes brain network segregation as a general feature of processing efficiency not only in feedback-driven learning as in the present study but also in other types of learning and in other task domains.’

In addition, we also slightly modified the Discussion (page 25, line 446-452) to further reduce ambiguity and thereby to avoid confusion: ‘More generally speaking, our present findings are in line with the notion that increasing network segregation might be a common consequence of more efficient task performance not only in feedback-driven learning but also in different types of learning and task domains other than learning. An important next step for future studies will be to examine the more fine-grained structure of these large-scale network changes in order to disentangle potential domain-specific and domain-general aspects associated with differences in task efficiency.’

2. Including a schematic of the task design would greatly help the reader in grasping specifics of the task.

We thank the reviewer for this suggestion.

We have added a Figure from the original Zwosta et al. 2018 paper. Note that for the dynamic connectivity analysis only data from phase 2 were used. The purpose of phase 1 and phase 3 was to generate the behavioral index of ‘habit strength’.

Figure 1. Experimental paradigm for study 1 with three consecutive phases (A, B, C). In study 2 the number of stimuli was reduced by 50% and only the avoidance learning condition was implemented in phase 2. For the dynamic functional brain state analysis only data from phase 2 were used. The purpose of phase 1 and phase 3 was to generate a behavioral index of ‘habit strength’. (A) Examples of the instructed hierarchical associations between stimulus category, response, and outcome (background color) used in phase 1 and two exemplary trials from phase 1. In phase 1 goal-directed behavior was established: given a certain stimulus category (artificial or natural), participants had to execute a certain action (left or right key press) in order to change or maintain the background color (no stimulus-specific associations could be learned). (B) Exemplary trials from Phase 2. In phase 2 participants had to learn by trial-and-error stimulus-specific associations with the two response options (‘habits’). In the approach condition a correct response was indicated by monetary gain (study 1 only). In the avoidance condition a correct response was indicated by the absence of monetary loss (study 1 and study 2). (C) Examples for compatible, incompatible trials in Phase 3. In phase 3 goal-directed responses established in phase 1 were put into competition with responses trained in phase 2 in order to probe individual habit strength.

3. Results Pg 10/11: "We found a significant negative correlation between the learning rate and slope of the modularity-Q...subjects with a higher learning rate were the subjects who were faster in transition from a more integrated into a more segregated brain state therefore showing less increase in segregation across training". The negative direction of this behavioral association is counterintuitive, and could do with some more unpacking. The analyses in Fig 3D/E that break down the modularity timecourses by fast and slow learners are helpful. These suggest that the key difference in network segregation between these subject groups appears **early** in learning, as increased segregation for fast learners (especially for Study 2, Fig 3E), which likely underpins the slope differences. Could the authors target this early effect more focally e.g. correlate the early modularity values (perhaps averaging them across block 1-3) with behavioral learning rate and habit strength? This might yield a more intuitive positive association with learning rate, and complement the current analyses.

We appreciate this insightful question.

As the reviewer suggested, we averaged the sliding windows 1 to 3 (‘early modularity values’) and correlated it with both learning rate and habit strength. There was a significant correlation between the early modularity values and learning rate in study 2 ($r = 0.2961$, $p = 0.0019$, one-tailed) and also for study 1 ($r = 0.2548$, $p = 0.0355$, one-tailed). However, and again as in the original analysis, we did not find a significant correlation between the early modularity values and habit strength neither in study 1 ($r = 0.1192$, $p = 0.4049$) nor study 2 ($r = 0.0151$, $p = 0.8853$).

We added this complementary analysis on page 12, line 219-228: ‘As suggested by visual inspection of **Fig, 4d and 4e**, network segregation seems to happened earlier for quicker learners (and for the easier task, see **Fig 4a**). To confirm this statistically, we averaged ‘early modularity values’ from sliding windows 1 to 3 and correlated it

with both learning rate and habit strength. There was a significant correlation between the early modularity values and learning rate in study 2 ($r = 0.2961$, $p = 0.0019$, one-tailed) and also in study 1 ($r = 0.2548$, $p = 0.0355$ one-tailed). However, and again as in the primary analysis, we did not find a significant correlation between the early modularity values and habit strength neither in study 1 ($r = 0.1192$, $p = 0.4049$) nor study 2 ($r = 0.0151$, $p = 0.8853$).

4a. The authors should better justify why learning rate and habit strength are treated as independent processes in their analysis. Intuitively, the extent to which task rules are learned more rapidly should lead to quicker automation (possibly involving rule consolidation), and hence associate with stronger habit formation; indeed, is there a correlation between the two behavioral measures? Better articulation of how learning rate differs process-wise from habit strength in the introduction might also better convey the novel contribution of this paper versus previous work (e.g. the Basset et al., 2015, NatNeuro paper).

We thank the reviewer for raising this excellent question.

In our study learning rate and habit strength were uncorrelated both in study 1 ($r = 0.0149$, $p = 0.9172$) and in study 2 ($r = 0.054$, $p = 0.6049$). We apologize for omitting this important piece of information and have now added it in the main text (page 23, after line 386).

Accordingly, we argued in the main text on page 5 in line 88-91: ‘...learning rate indicating how quickly individual subjects are able to extract the new rules by trial-and-error while habit strength was considered as a measurement of how well the newly acquired rules were being automatized through extended practice or, in other words, how enduring the trained S-R associations are.’

Note that this interpretation is nicely supported by the results of the additional analysis focusing on the early learning blocks as requested by the reviewer (see our reply to the previous point).

Moreover, this question might also connect with the next question raised by the reviewer. In our paradigm, due to a relatively short training duration, learning rate likely predominantly reflects progress in early rule extraction rather than progress in habitualization or automatization. In other paradigms the focus is more on the effects of very extensive training and even without an initial trial-and-error stage (e.g. Basset et al. 2015). Thus, under such circumstances learning rate (measured by RT decrease in that case) likely reflects increasing habitualization. In turn, this is probably the reason why in our case, learning rate and habit strength are uncorrelated. We added a few sentences on page 23 after line 386: ‘We also failed to find a significant correlation between learning rate and habit strength ($r = 0.0149$, $p = 0.9172$ in study 1 and $r = 0.054$, $p = 0.6049$ in study 2). Together with the significant correlation results

between early modularity values and learning rate, this again suggests that learning rate indicates how quickly individual subjects are able to extract the new rules by trial-and-error. This contrasts with the absence of a significant correlation between early modularity values and habit strength as a measure of how well the newly acquired rules were being automatized through extended practice or, in other words, how enduring the trained S-R associations are.'

4b. In interpreting the observed lack of behavioral association between network segregation and habit strength, the authors correctly highlight the possibility that their brain measures are not sensitive enough (Pg 21): "Importantly, this does of course not exclude that other indices of brain activity or connectivity dynamics might be associated with habit-related processes 13,14,51". This important point would be strengthened by citing relevant recent work looking at multivariate/representational brain measures of learning (Mill & Cole (2023), "Neural representation dynamics reveal computational principles of cognitive task learning"; Tambini et al (2023): "Structured memory representations develop at multiple time scales in hippocampal-cortical networks"). Evaluating this work might also inform a necessary discussion as to whether neural associations with habit strength were not obtained in the present report due to looking at a relatively short window of learning. Stronger associations with habit strength would perhaps emerge from more extended task learning and consolidation.

We thank the reviewer for pointing out these two recent papers.

As the reviewer suggested, we have integrated those latest references into the revised version (page 23 after line 399).

'Importantly, this does of course not exclude the possibility that other neural indices might be associated with behavioral indices of habit formation. For example, previous research found that training-related brain activity changes in angular gyrus (Zwosta et al., 2018) and head of caudate (Gera et al., 2023) were associated with habit strength. Connectivity-wise, from our previous research, we found that the functional connectivity changes involving the sensorimotor and the cingulo–opercular network contributed most prominently to habit strength prediction (Wang et al., 2022). In addition, Mill et al. found that the geometry of neural representations changes, originating in subcortex (hippocampus and cerebellum) and slowly spreading to cortex, to support a transition from novice to practiced performance (Mill et al., 2023). Moreover, by applying representational similarity analysis, Tambini et al. found that the specific hippocampal representations emerge early, followed by both specific and schematic representations at a gradient of timescales across hippocampal-cortical networks during the longitudinal memory representation task (Tambini et al., 2023).'

We also totally agree with the reviewer that one possible reason why we did not find a significant correlation between habit strength and network segregation

dynamics might be the relatively short overall training duration (accordingly accompanied by a shorter sliding window) compared to other studies (Basset et al., 2015). Please, see our reply to the preceding comment for further elaboration.

5. I had some difficulty in interpreting the FPN learning dynamics. Pg 19: "The FPN, which is considered to comprise flexible hub regions involved in high-level cognitive control 42-44, demonstrated decreasing modularity across learning in the present study. These results are consistent with previous studies which demonstrated a positive correlation between FPN activation and S–R rule difficulty 45 as well as between the within-network functional connectivity strength of the FPN and working memory load 46." This needs to be unpacked more clearly - are the authors suggesting that the FPN is somehow tracking the decrease in task difficulty over learning? Still, this does not explain why this neural tracking would take the form of a decrease in network segregation per se (versus e.g. a first-order decrease in network activation). I think it would help to provide results for FPN (and perhaps DMN, which the authors also focus on) in the same format as Figure 3D/E, and (per my point 3 above) also interrogate whether the FPN modularity slope effects are driven by an early increase in modularity for slow learners (who experience the task as more difficult than fast learners). This would provide some support for the authors' hypothesized tracking of task difficulty by the FPN's segregation.

We appreciate this insightful question.

Indeed, as the reviewer commented, compared with a first-order decrease in FPN activation, the decreased Module-degree Z (MDZ) of FPN reported in the present paper might not be straightforward. However, this decreased MDZ across training resembles previous study results (Repovs et al., 2012). Based on the analysis of within-module functional connectivity (as a partial measure of module segregation; the higher the more segregated) they showed smaller within-module connectivity when working memory load/task difficulty is smaller. This might suggest that simpler tasks require less coordination and communication within the FPN. Applied to the present study, decreasing task demand due to progress in learning would go along with reduced communication within the FPN as reflected by decreasing MDZ (in this case mainly driven by decreasing within-FPN connectivity).

We added more information on page 21, line 361: 'These results are consistent with previous studies which demonstrated a positive correlation between FPN activation and S–R rule difficulty (Woolgar et al., 2015) as well as between the within-network functional connectivity strength of the FPN and working memory load (Repovs et al., 2012). In Repovs et al. the within-module functional connectivity was lower for lower working memory load/task difficulty, which might suggest that simpler tasks require less coordination and communication within the FPN. Applied to the present study results, this implies that the learning-related decrease in FPN segregation reflects reduced requirement for communication within the FPN due to reduced cognitive

control demands when the task becomes more automatized.’

In the original main text, **Figure 3d/e** (comparing the time courses of the modularity-Q value for different groups of subjects defined by a median split according to individual learning rate) was created to provide a more easily interpretable representation of the correlation between the learning rate and dynamics of network segregation. However, we did not find a significant correlation between the decreasing FPN-related segregation and learning rate in the first place (same for habit strength). Hence, contrary to the reviewer’s suggestion, we think that it does not make sense to perform follow-up analyses if the primary analysis failed significance. However, in contrast to the FPN, for the DMN we did find a significant negative correlation between increased segregation and learning rate in both study 1 ($r = -0.2959$, $p = 0.0390$) and study 2 ($r = -0.2234$, $p = 0.0314$). We therefore implemented the suggested follow-up analyses for the DMN and plotted the results for the DMN in the same format as in original **Figure 4d/e**. We also computed the correlation results between the early values (averaging across the first 3 blocks) and learning rate and habit strength. Regarding the dynamic changes in MDZ and PC scores for DMN, the two studies demonstrated the same pattern: an early MDZ increase paralleled by a decreased PC score for the higher learning rate group. There was also a significant correlation between learning rate and both the early MDZ ($r = 0.2544$, $p = 0.0134$) as well as early PC ($r = -0.2238$, $p = 0.0301$) in study 2. In study 1 these correlations were found to be in the same direction as in study 2, but unfortunately failed significance ($r = 0.1952$, $p = 0.1699$ for MDZ and $r = -0.1073$, $p = 0.4535$ for PC). Again, no significant correlation was found between the neural index and habit strength in both studies.

In our current study, the general approach was to report detailed results only when replicated across both studies. Thereby we aimed to establish robust effects in trial-and-error learning that generalize across task specifics. We therefore decided to only report these DMN-related results in this response letter for transparency, but would like to keep it out of the manuscript.

Changes in mean Module-degree Z (MDZ) and Participation Coefficient (PC) in DMN for high learning rate and low learning rate subgroups for study 1 and study 2.

Minor concerns:

1. Intro pg 3: "With increasing practice, however, behavior is assumed to become less and less governed by outcome anticipation and instead to become more and more controlled directly by stimulus-response (S-R) associations". The use of "control" here is confusing as it connotes "cognitive control", which could be argued to decrease as learning progresses towards automaticity. Please consider rewording.

We changed this to 'With increasing practice, however, behavior is assumed to become less and less governed by outcome anticipation and instead becomes more and more driven by stimulus-response (S-R) associations.' on page 3 after line 51.

2. To help the reader, Figure 2 should include labels linking state 1 to network integration and state 2 to network segregation.

The habit strength results on pg 11 should mention how to interpret the sign of habit strength i.e. positive values = stronger habit formation.

We have modified the figure as the reviewer suggested on page 11.

We also added the description to clarify that a higher compatibility effect indicated stronger habit strength on page 13, line 232: ‘...for similar results based on habit strength defined by the RT-based compatibility effect (higher compatibility effect indicated stronger habit strength).’

3. Results pg 13: please include some justification for transitioning from using modularity alone for the preceding behavioral analysis to using modularity and participation coefficient for these individual network results.

In order to follow the reviewers request, in the main text we added the following information on page 38 after line 716 and page 39 after line 747: ‘The modularity Q value is a widely used index when considering the whole brain as a globally interconnected modular system. The participation coefficient and the Module-Degree Z score are complementary indices that allow for more fine-grained conclusions regarding individual brain networks/modules. We followed a previous suggestion to

combine Module-Degree Z score and participation coefficient into a two-dimensional plot or cartographic profile (Shine et al., 2016). ‘

Reviewer #3 (Remarks to the Author):

The authors examine the relationship between large scale brain network dynamics and learning. They report that a faster learning rate is associated with a faster shift to network segregation relative to network integration. This relationship was evident with learning rate, but not habit strength. The authors identify similar results in two different datasets and clearly describe their scientific process from the descriptions of data analysis to their interpretation of the results. The manuscript is well written. I have a few suggestions that may improve the paper.

I could not find a description of the duration of the sliding windows used or the overall duration of the task phase.

We thank the reviewer for reminding us and apologize for omitting this relevant information in the original submission.

We added this information on page 35, line 660-668: ‘The duration of the sliding windows and overall duration is slightly different among subjects depending on their individual response speed. There were 784 and 392 learning trials in study 1 and study 2. The average number of scans across subjects are 1366.92 in study 1 and 655.09 in study 2, since the whole task was divided into 20 non-overlapping windows in study 1 and 20 overlapping windows with half window step in study 2. We chose overlapping windows in study 2 to obtain the same number of windows as well as duration of windows as in study 1 considering that time series were only half as long as in study 1. The average duration of the sliding window is 68.35 scans in study 1 and 65.51 scans in study 2.’

The authors cite Cole et al., 2019 for the approach of removing mean task activation and calculating functional connectivity from the residuals of said model. However, Cole et al. also found that FIR is more effective at removing the possible inflating influence of task co-activation from estimates of connectivity. The authors here use a canonical HRF in their task model. The authors should consider using a FIR approach or acknowledging that their estimates of connectivity may be inflated to some degree.

We thank the reviewer for this valuable comment and we acknowledge that the method for regressing average task-related activity was limited by using the canonical HRF basis function instead of the Finite Impulse Response (FIR) basis functions. On page 25, line 438-445, we add more information to further acknowledge this limitation. ‘There is still one limitation we should bear in mind when computing the functional connectivity, that is, the strategy for regressing out average task-related activity. Cole et al. (2019) found that Finite Impulse Response (FIR) basis functions might be a better choice than others for removing the potential influence of mean task-related co-activations from estimates of connectivity (Cole et al., 2019). Further

studies might consider using FIR or other more flexible basis functions like Fourier sets as the primary method for regressing the mean task activity when computing the task-based functional connectivity.’

One major question I was left with after reading the manuscript was: Are the network shifts to segregation actually related to learning, or is there another possible explanation? The authors correlate these shifts in network topology to individual differences in learning which provides some support for the authors’ interpretation. However, I wonder if the same shifts in network topology would be observed during rest or during the performance of an unrelated task. Is it possible that network segregation increases as scans get longer in general? Could the changes in network topology be related to participants spending less “time on task” as the experiment progresses? Or could the start of a scan be related to increased arousal/vigilance, which wanes over time much like the shift from integration to segregation? The manuscript would be stronger if the authors could address some of these questions and provide additional support for the conclusion that network topology reflects learning rather than other possible explanations.

We are grateful to the reviewer for bringing this up.

Based on the reviewers question we included a new paragraph, where we explicitly discuss the issue on page 24, line 413-437: ‘We believe there are good reasons to argue that increased segregation observed in our study mainly reflects learning-related changes is task difficulty/automatization rather than more unspecific effects merely related to the passage of time as would be the case during resting state studies. Specifically, brain state transitions in resting state studies seem to follow different patterns than we observed in the present study. One previous study found that the prevalence of segregated vs. integrated states during resting was fluctuating over time but there was no systematic trend in one or the other direction like in our study (Shine et al., 2016). Moreover, another study found that segregation actually decreased across resting state scanning (Allen et al., 2012). This is the opposite direction than in the present study and in other learning studies (Bassett et al., 2015; Mohr et al., 2016). Moreover, a resting state study which involved sleep deprivation demonstrated that different arousal states were indeed associated with different brain states (Wang et al., 2015). However, this association was opposite to what would be expected if our own present results were driven by hypothetical changes in arousal being confounded with progress in learning (i.e., high arousal in the beginning associated with more integrated state vs. low arousal in the end associated with more segregated state). Specifically, in Wang et al. (2015), the low arousal state was associated with a more ‘integrated’ brain state with reduced within network functional connectivity of default mode and dorsal/ventral attention networks, as well as reduced anticorrelation between these networks. In contrast, the high arousal state was associated with the more ‘segregated’ brain state with more decoupling between the default mode

networks and higher-order cognitive networks that included dorsal/ventral attention networks (Wang et al., 2015). To conclude, in light of of previously published resting state studies (Shine et al., 2016; Allen et al., 2012; Wang et al., 2015) as well as previous studies on learning (Bassett et al., 2015; Mohr et al., 2016), our present observation of increased network segregation across learning seems most likely to be related to progress in learning, not least suggested by the significant association with learning rate.'

Reference:

- Allen, E. A. et al. Tracking whole-brain connectivity dynamics in the resting state. *Cereb Cortex* 24, 663-676, doi:10.1093/cercor/bhs352 (2014).
- Bassett, D. S. et al. Dynamic reconfiguration of human brain networks during learning. *Proceedings of the National Academy of Sciences of the United States of America* 108, 7641-7646, doi:10.1073/pnas.1018985108 (2011).
- Cole, M. W. et al. Task activations produce spurious but systematic inflation of task functional connectivity estimates. *Neuroimage* 189, 1-18, doi:10.1016/j.neuroimage.2018.12.054 (2019).
- Gera, R. et al. Characterizing habit learning in the human brain at the individual and group levels: a multi-modal MRI study. *NeuroImage* 272, 120002, doi:https://doi.org/10.1016/j.neuroimage.2023.120002 (2023).
- Liao W, Wu GR, Xu Q, Ji GJ, Zhang Z, Zang YF, Lu G. DynamicBC: a MATLAB toolbox for dynamic brain connectome analysis. *Brain Connect.* Dec;4(10):780-90. doi: 10.1089/brain.2014.0253. PMID: 25083734; PMCID: PMC4268585 (2014).
- Mill RD, Cole MW. Neural representation dynamics reveal computational principles of cognitive task learning. *bioRxiv [Preprint]*. Jun 28:2023.06.27.546751. doi: 10.1101/2023.06.27.546751. PMID: 37425922; PMCID: PMC10327096 (2023).
- Mohr, H. et al. Integration and segregation of large-scale brain networks during short-term task automatization. *Nat Commun* 7, 13217, doi:10.1038/ncomms13217 (2016).
- Newman, M. E. J. Fast algorithm for detecting community structure in networks. *PHYSICAL REVIEW* 69, 066133 (2004).
- Repovs, G. & Barch, D. M. Working memory related brain network connectivity in individuals with schizophrenia and their siblings. *Frontiers in human neuroscience* 6, 137, doi:10.3389/fnhum.2012.00137 (2012).
- Rubinov, M. & Sporns, O. Complex network measures of brain connectivity: uses and interpretations. *Neuroimage* 52, 1059-1069, doi:10.1016/j.neuroimage.2009.10.003 (2010).
- Shine, James M. et al. The Dynamics of Functional Brain Networks: Integrated Network States during Cognitive Task Performance. *Neuron* 92, 544-554, doi:https://doi.org/10.1016/j.neuron.2016.09.018 (2016).
- Tambini A, Miller J, Ehlert L, Kiyonaga A, D'Esposito M. Structured memory representations develop at multiple time scales in hippocampal-cortical networks. *bioRxiv [Preprint]*. Apr 7:2023.04.06.535935. doi: 10.1101/2023.04.06.535935. PMID: 37066263; PMCID: PMC10104124 (2023).
- Vatansever, D., Menon, D. K., Manktelow, A. E., Sahakian, B. J. & Stamatakis, E. A. Default Mode Dynamics for Global Functional Integration. *The Journal of neuroscience : the official journal of the Society for Neuroscience* 35, 15254-15262, doi:10.1523/jneurosci.2135-15.2015 (2015).
- Wang C, Ong JL, Patanaik A, Zhou J, Chee MW. Spontaneous eyelid closures link vigilance fluctuation with fMRI dynamic connectivity states. *Proc Natl Acad Sci U S A.* Aug 23;113(34):9653-8. doi: 10.1073/pnas.1523980113. Epub 2016 Aug 10. PMID: 27512040; PMCID: PMC5003283 (2016).
- Wang, X., Zwosta, K., Wolfensteller, U. & Ruge, H. Changes in global functional network properties predict individual differences in habit formation. *Hum Brain Mapp*, doi:10.1002/hbm.26158 (2022).
- Woolgar, A., Afshar, S., Williams, M. A. & Rich, A. N. Flexible Coding of Task Rules in Frontoparietal

Cortex: An Adaptive System for Flexible Cognitive Control. *Journal of cognitive neuroscience* 27, 1895-1911, doi:10.1162/jocn_a_00827 (2015).

Zwosta, K., Ruge, H., Goschke, T. & Wolfensteller, U. Habit strength is predicted by activity dynamics in goal-directed brain systems during training. *Neuroimage* 165, 125-137, doi:10.1016/j.neuroimage.2017.09.062 (2018).

REVIEWERS' COMMENTS:

Reviewer #1 (Remarks to the Author):

The authors have extensively addressed all my concerns. I therefore suggest publication.
Best regards

Reviewer #2 (Remarks to the Author):

I am satisfied with the authors' revisions, and commend them on their insightful report. I would now recommend acceptance for publication.

Reviewer #3 (Remarks to the Author):

This is a review for a revised manuscript. The authors have addressed feedback from the reviewers and as a result, strengthened the manuscript. I do not have any additional feedback and would support accepting the manuscript for publication.